# Regulation of the immune tolerance system determines the susceptibility to HLA-mediated abacavir-induced skin toxicity

Takeshi Susukida [1,2,4], Saki Kuwahara[1,4], Binbin Song[3], Akira Kazaoka[1], Shigeki Aoki [1✉] & Kousei Ito[1]

Idiosyncratic drug toxicity (IDT) associated with specific human leukocyte antigen (HLA) allotype is a rare and unpredictable life-threatening adverse drug reaction for which prospective mechanistic studies in humans are difficult. Here, we show the importance of immune tolerance for IDT onset and determine whether it is susceptible to a common IDT, HLA-B*57:01-mediated abacavir (ABC)-induced hypersensitivity (AHS), using CD4+ T cell-depleted programmed death-1 receptor (PD-1)-deficient HLA-B*57:01 transgenic mice (B*57:01-Tg/PD-1$^{-/-}$). Although AHS is not observed in B*57:01-Tg mice, ABC treatment increases the proportion of cytokine- and cytolytic granule-secreting effector memory CD8+ T cells in CD4+ T cell-depleted B*57:01-Tg/PD-1$^{-/-}$ mice, thereby inducing skin toxicity with CD8+ T cell infiltration, mimicking AHS. Our results demonstrate that individual differences in the immune tolerance system, including PD-1$^{high}$CD8+ T cells and regulatory CD4+ T cells, may affect the susceptibility of humans to HLA-mediated IDT in humans.

[1] Laboratory of Biopharmaceutics, Graduate School of Pharmaceutical Sciences, Chiba University, Chiba, Japan. [2] Laboratory of Cancer Biology and Immunology, Section of Host Defences, Institute of Natural Medicine, University of Toyama, Toyama, Japan. [3] Key Laboratory of Ethnomedicine (Minzu University of China), Ministry of Education, School of Pharmacy, Minzu University of China, 100081 Beijing, China. [4] These authors contributed equally: Takeshi Susukida, Saki Kuwahara. ✉email: aokishigeki@chiba-u.jp

diosyncratic drug toxicity (IDT) is an unpredictable, potentially life-threatening adverse event, as it is difficult to perform prospective mechanistic studies in humans. IDT-related symptoms can occur ubiquitously throughout the body, mostly manifesting as a skin rash, Stevens–Johnson syndrome (SJS)/toxic epidermal necrolysis (TEN), drug reaction with eosinophilia and systemic symptoms (DRESS), and drug-induced liver injury (DILI). Their unpredictable nature makes prospective mechanistic studies in humans difficult, and therefore, our knowledge of IDT remains limited[1]. Adverse drug effects, including IDT, are the fourth to sixth leading causes of death in the United States while causing ~197,000 deaths annually in Europe[2]. This indicates that IDT should not be underestimated in clinical practice. Thus, there is a need to establish a proper evaluation method to investigate the underlying mechanism of IDT and identify potential IDT of candidate compounds during preclinical development, in order to reduce the risk of marketing drugs with hazardous properties and improve safety in patients.

The involvement of the adaptive immune system in the pathogenesis of IDT is indicated by common clinical characteristics, including the presence of human leukocyte antigen (HLA) risk alleles and drug-induced proliferation of lymphocytes isolated from patients[3]. Specifically, several studies have provided evidence that the IDT onset risk is associated with specific HLA haplotypes[4,5]. For instance, the strong association between abacavir (ABC) hypersensitivity (AHS) manifesting as a skin rash and the HLA-B*57:01 allele is reportedly based on a high odds ratio (>900)[6–9]. AHS is induced by abnormal activation of CD8+ T cells with the concomitant release of Th1 cytokines (i.e., interferon-gamma (IFN-γ) and interleukin-2 (IL-2))[10–13]. Furthermore, flucloxacillin (FLUX)-associated DILI has been observed in HLA-B*57:01 carriers[14]. However, IDT does not occur in all subjects of a given HLA model population. In fact, inadequate positive prediction values of each HLA-associated IDT have been reported as follows: AHS (a positive result in epicutaneous patch test) in HLA-B*57:01, 47.9%[15]; allopurinol-induced SJS/TEN in HLA-B*58:01, 76.5%[16]; carbamazepine-induced SJS/TEN in HLA-B*15:02, 5.6%[17]; and carbamazepine-induced DRESS in HLA-A*31:01, 0.59%[18]. These findings suggest the difficulties in attempting to predict the IDT risk during preclinical studies without considering additional factors besides genetic HLA factors.

HLA transgenic (Tg) mouse models represent a valuable tool to prospectively investigate the mechanisms underlying HLA-mediated autoimmune diseases and IDTs[19]. For instance, a Tg mouse line carrying the human-mouse chimeric HLA-B*57:01 (B*57:01-Tg) can be used to evoke ABC-induced abnormal immune toxicity[20,21]. However, these studies have neither reproduced AHS symptoms observed in clinical situations nor investigated the underlying mechanism of AHS, implying that a mouse model solely based on the introduction of this is not sufficient to comprehensively understand the underlying mechanism of HLA-related IDT. Therefore, it is imperative to identify factors that determine the susceptibility to IDT using HLA Tg IDT models.

Here, we examined whether immune tolerance contributes to the susceptibility to AHS induced by oral ABC in an HLA-IDT mouse model. The study revealed an unclarified onset risk factor of IDTs using an appropriate HLA Tg mouse model and focused on AHS in HLA-B*57:01 mice. As the majority of AHS episodes in terms of high odds ratio occurs in patients taking ABC who present human immunodeficiency virus (HIV)-induced low CD4+ T cell count, including regulatory T cells (Treg)[22–24] specialized in immune suppression[25], the excessive activation of CD8+ T cells observed in HIV-infected patients[23,26] seems to be essential to induce AHS. In fact, a previous study suggested the importance of depleting CD4+ T cells to exacerbate ABC-induced abnormal immune toxicity by treating another B*57:01-Tg mice line with both i.p. injection and ear painting, possibly by enhancing antigen-presenting cell co-stimulation[21]. Thus, we hypothesized that impairing immune tolerance in B*57:01-Tg mice would reflect the clinical situation of patients on ABC, leading to a promising AHS evaluation model, which could provide insights into the events and processes leading to IDT. Furthermore, it is important to reproduce IDTs in an HLA Tg mouse model using oral drug administration, as IDTs are mostly associated with oral prescription medications[27].

## Results

**Skin toxicity on the ear in CD4+ T cell-depleted B*57:01-Tg mice treated with oral ABC for 3 weeks.** Because the majority of patients taking ABC are HIV-positive and thus have a low CD4+ T cell count[23,24], we initially examined whether depleting CD4+ T cells, including Treg, caused ABC-induced skin rash in B*57:01-Tg mice. To enable a clear assessment of topical skin toxicity, the mice were fed ABC-containing rodent chow containing ABC (1% w/w) plus ear painted (50 mg/kg/day) and administered either anti-mouse CD4 monoclonal antibody (mAb) (0.25 mg/body) or vehicle (PBS) by i.p. injection for 3 weeks. This was a modified protocol of another toxicological study using B*57:01-Tg mice, treated with both i.p. injection and ear painting of ABC[21]. On day 21, B*57:01-Tg mice treated with anti-CD4 mAb plus ABC showed robust signs (i.e., redness and congestion) of skin hypersensitivity associated with the CD8+ T cell infiltration of the dermic layer on both ears (Supplementary Fig. 1). Consistent with our previous study finding[20], the skin hypersensitivity was not observed even in CD4+ T cell-depleted ABC-fed B*57:03-Tg mice (Supplementary Fig. 1), which possess the closely related, non-AHS-associated HLA allotype control that ABC cannot specifically react because of a two amino acid difference at the binding site[10].

In clinical practice, ABC 300 mg twice daily or 600 mg once daily is orally administered as a tablet or solution[28]. Therefore, we examined whether the conspicuous skin toxicity could also be induced without ear painting in CD4+ T cell-depleted B*57:01-Tg mice receiving oral 1% (w/w) ABC (Fig. 1). This dose of ABC could sufficiently induce abnormal CD8+ T cell activation in the lymph node (LN) and spleen in B*57:01-Tg mice by maintaining an average ABC plasma concentration of 34.5 μM, which is ~3-fold higher than the maximum plasma concentrations ($C_{max}$) in humans[20,29]. Lymphocytic and CD8+ T cell dermal infiltration in the ear was only observed in the CD4+ T cell-depleted ABC-fed B*57:01-Tg mice (Fig. 1a), which displayed a significant increase in the percentage of effector memory CD8+ T cells (CD44highCD62Llow) in the auricular LN ($p < 0.05$) (Fig. 1b and c). These signs of skin hypersensitivity on the ears were observed on day 21 of oral ABC administration irrespective of whether the ears were painted or not. However, this result differed from clinical cases in humans with a typical median time to AHS onset of 11 days[30]. Moreover, signs of obvious ear inflammation (e.g., redness or congestion) were not observed in the oral ABC administrated CD4+ T cell-depleted B*57:01-Tg mice (Fig. 1a). Hence, we considered the existence of another AHS suppressor in the CD4+ T cell-depleted B*57:01-Tg mice.

**PD-1 expression level in CD4+ T cell-depleted B*57:01-Tg mice treated with ABC for 1 week.** The immune tolerance system mainly consists of two immune checkpoints, a programmed cell death protein 1 (PD-1) on T cells and the cytotoxic T lymphocyte-associated antigen 4 (CTLA-4) on CD4+ Treg[31]. As AHS is mediated by CD8+ T cells, we focused on CD8+ T cell

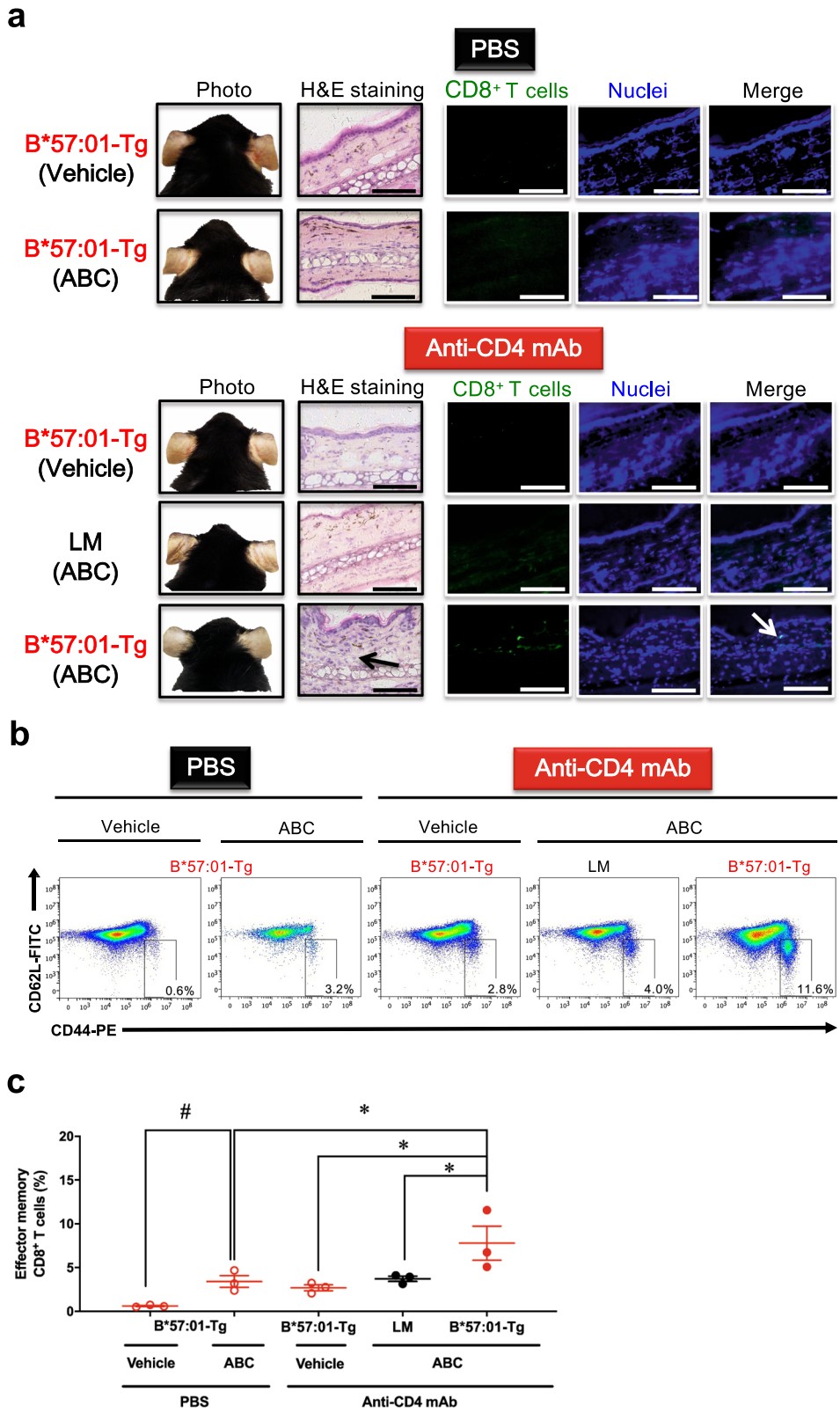

PD-1 as an AHS suppressor in CD4+ T cell-depleted B*57:01-Tg mice. To determine whether PD-1 contributes to the immuno-suppression in the CD4+ T cell-depleted scenario, we measured PD-1 expression of effector memory CD8+ T cells in CD4+ T cell-depleted B*57:01-Tg mice (Fig. 2). Upregulated PD-1 expression was observed on day 7 in both auricular LNs and spleen-derived effector memory CD8+ T cells of ABC-fed B*57:01-Tg mice, compared with that in littermate (LM) mice, and its expression was further upregulated in effector memory CD8+ T cells of CD4+ T cell-depleted B*57:01-Tg mice (Fig. 2a). Under CD4+ T cell depletion, the effector memory CD8+ T cells in the LNs of ABC-fed B*57:01-Tg mice presented significantly higher PD-1 expression levels than those in the LMs and B*57:03-Tg mice (Fig. 2b). Moreover, these PD-1 levels were also

**Fig. 1 Effect of 3-week oral abacavir (ABC) administration on the ear in CD4$^+$ T cell-depleted HLA-B*57:01 transgenic (Tg) mice. a** Representative images of the ears of B*57:01-Tg mice and their littermates (LMs) on day 21. The ear sections were either stained with hematoxylin and eosin (H&E) or immunostained with anti-CD8, along with Hoechst 33342 nuclear staining. Both groups received oral ABC (1% w/w) or a normal diet (vehicle) for 3 weeks, with i.p. injections of 0.25 mg anti-CD4 mAb or PBS. Data are representative of three independent experiments. Each scale bar represents 100 μm. Arrows mark either the lymphocytic infiltration (H&E staining) or CD8$^+$ T cell infiltration (CD8 immunohistochemistry). **b** Representative dot plots depicting effector memory T cells in gated CD8$^+$ T cells classified by the phenotype of CD44 and CD62L expression in the lymph nodes (LNs) and **c** percentage of effector memory T cells among CD8$^+$ T cells isolated from the auricular LN of mice. Effector memory CD8$^+$ T cells were gated from lymphocytes by anti-CD44 antibody and anti-CD62L antibody (phenotype: CD44$^{high}$CD62L$^{low}$). Each plot represents an individual mouse with the mean ± SEM ($N = 3$); *$p < 0.05$, compared with other B*57:01-Tg mouse groups or LM mice, #$p < 0.05$, compared with PBS-treated vehicle-fed B*57:01-Tg mice; one-way ANOVA with Bonferroni's multiple comparisons correction. Data are **b** representative of three independent experiments or **c** summary of three independent experiments.

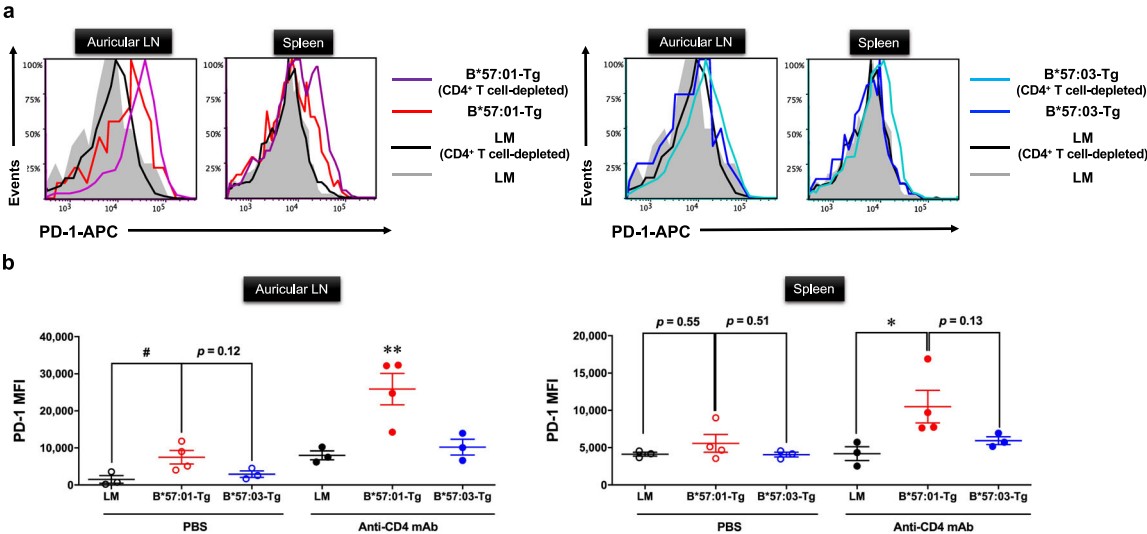

**Fig. 2 PD-1 surface expression of effector memory CD8$^+$ T cells in CD4$^+$ T cell-depleted HLA-transgenic (Tg) mice. a** Flow cytometric measurement of PD-1 surface expression. Representative histograms of PD-1 surface expression on effector memory CD8$^+$ T cells in the auricular lymph node (LN) or spleen from CD4$^+$ T cell-depleted/PBS-treated B*57:01-Tg mice (left panel), B*57:03-Tg mice (right panel), or their littermates (LMs). Mice have orally administrated 1% (w/w) abacavir (ABC) for 1 week. Effector memory CD8$^+$ T cells were gated from either lymphocytes or splenocytes by anti-CD44 and CD62L antibodies (phenotype: CD44$^{high}$CD62L$^{low}$). Data are representative of three independent experiments. **b** Median fluorescence intensity (MFI) values of PD-1 in effector memory CD8$^+$ T cells isolated from the auricular LN (left panel) or spleen (right panel) of 1% (w/w) ABC-fed mice for 1 week. Each plot represents an individual mouse with the mean ± SEM ($N = 3$–4); #$p < 0.05$, one-way ANOVA with Bonferroni's multiple comparisons correction, compared with PBS-treated mice groups; indicated $p$ values were obtained from a statistical comparison. *$p < 0.05$, **$p < 0.01$, one-way ANOVA with Bonferroni's multiple comparisons correction, compared with other mice groups. Data are a summary of three independent experiments.

significantly higher than those in control ABC-fed B*57:01-Tg mice. A significant upregulation of PD-1 expression was also observed in the spleen of ABC-fed B*57:01-Tg mice, compared with that in CD4$^+$ T cell-depleted LM mice (Fig. 2b). CD4$^+$ T cell-depleted vehicle-fed B*57:01-Tg mice did not show a significant increase in PD-1 expression on effector memory CD8$^+$ T cells ($p > 0.99$) (Supplementary Fig. 2).

**ABC-induced hypersensitivity in CD4$^+$ T cell-depleted B*57:01-Tg mice lacking PD-1.** We further examined whether the upregulation of PD-1 could suppress ABC-induced abnormal immune activation and skin hypersensitivity in B*57:01-Tg mice. For this purpose, we created a novel mouse line—PD-1 knockout HLA Tg mouse (B*57:01-Tg/PD-1$^{-/-}$ or B*57:03-Tg/PD-1$^{-/-}$ (negative control)). Initially, we assessed the percentage of effector memory CD8$^+$ T cells in 1-week ABC-fed mice (Fig. 3). CD4$^+$ T cell-depleted B*57:01-Tg mice that were fed ABC showed a substantial increase in the percentage of effector memory CD8$^+$ T cells in the LNs compared with that in the control B*57:01-Tg mice and the percentage further increased when PD-1$^{-/-}$ was introduced (Fig. 3a and b). The mean values of effector memory CD8$^+$ T cells in the LNs and spleen of CD4$^+$

T cell-depleted ABC-fed B*57:01-Tg mice were significantly higher than those in the respective LMs and B*57:03-Tg mice ($p < 0.05$) (Fig. 3c and Supplementary Fig. 3a); furthermore, the percentage of these effector memory CD8$^+$ T cells were significantly increased in B*57:01-Tg/PD-1$^{-/-}$ mice compared with that in other control ABC-fed mouse groups ($p < 0.001$), whereas the significant increase was not observed in PBS-treated B*57:01-Tg/PD-1$^{-/-}$ mice ($p > 0.12$). Vehicle (a normal diet) treatment of B*57:01-Tg/PD-1$^{-/-}$ mice did not significantly change the percentage of effector memory CD8$^+$ T cells ($p > 0.05$) (Supplementary Fig. 3b).

Next, we evaluated ABC-induced skin hypersensitivity in CD4$^+$ T cell-depleted B*57:01-Tg/PD-1$^{-/-}$ mice (Fig. 4). Consistent with the increased occurrence of effector memory CD8$^+$ T cells, robust signs of ear inflammation (Fig. 4a) and a systemic increase in LN size (Supplementary Fig. 4) were observed in the ABC-fed CD4$^+$ T cell-depleted B*57:01-Tg/PD-1$^{-/-}$ mice but not in other tested groups including the vehicle-fed counterparts. To assess whether the phenotypic features in CD4$^+$ T cell-depleted B*57:01-Tg/PD-1$^{-/-}$ mice were associated with CD8$^+$ T cell-mediated inflammation, the ear sections obtained after 1-week oral ABC treatment were subjected to either hematoxylin and eosin (H&E) staining or CD8 immunohistochemistry (IHC).

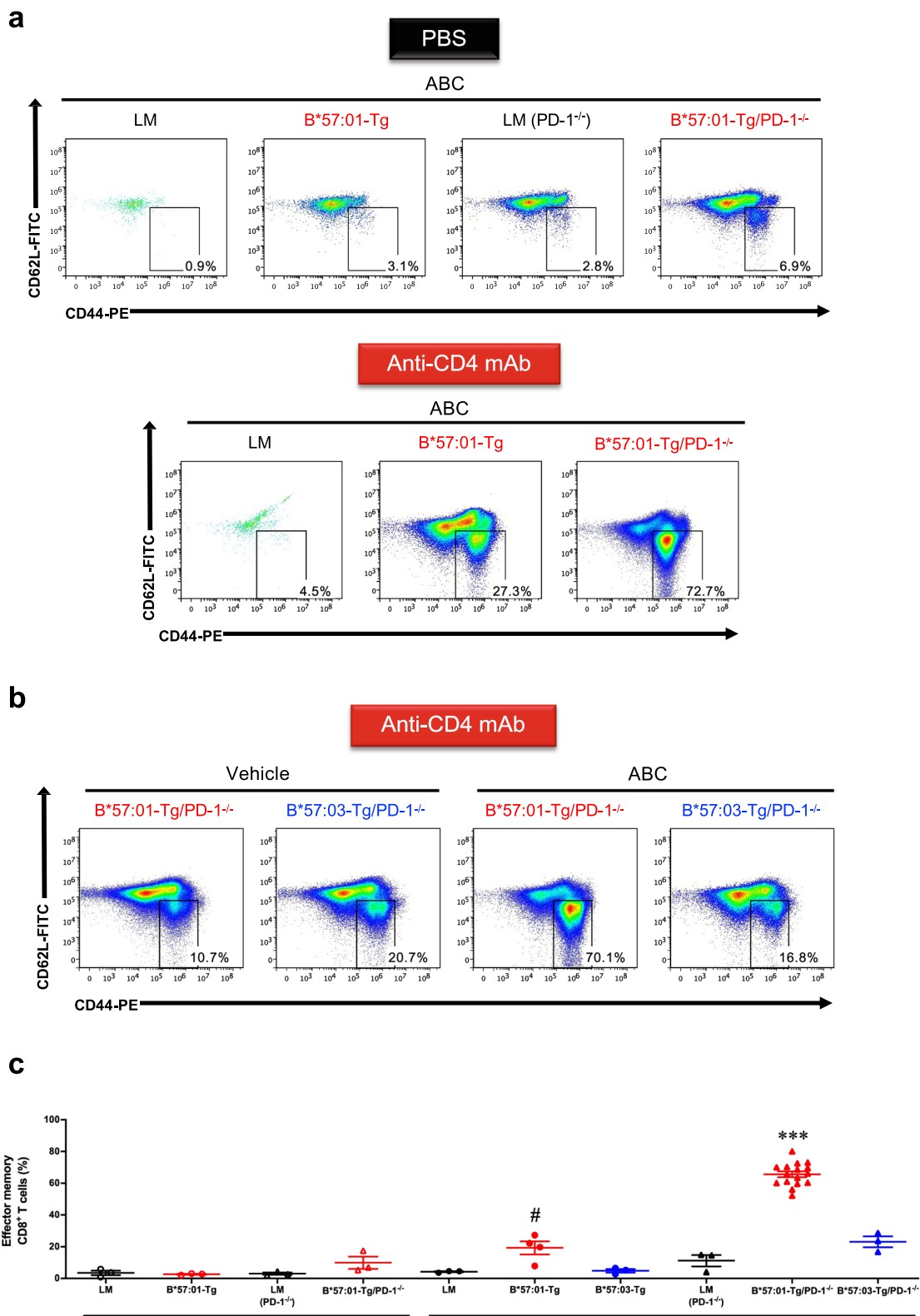

**Fig. 3 Effect of PD-1 knockout on abacavir (ABC)-induced activation of T cells in CD4$^+$ T cell-depleted HLA-B*57:01 transgenic (Tg) mice. a** and **b**
Representative dot plots depicting effector memory T cells in gated CD8$^+$ T cells, classified by the phenotype of CD44 and CD62L expression in the auricular lymph node (LN) from PBS-treated or CD4$^+$ T cell-depleted mice. Each group was orally administrated 1% (w/w) ABC or a normal diet (vehicle) for 1 week. Effector memory T cells: CD44$^{high}$CD62L$^{low}$. Data are representative of three independent experiments. **c** Percentages of effector memory T cells among CD8$^+$ T cells isolated from the auricular LN of 1% (w/w) ABC-fed mice. Each plot represents an individual mouse with the mean ± SEM ($N = 3$–16); \*\*\*$p < 0.001$, compared with other mice groups; #$p < 0.05$, compared with CD4$^+$ T cell-depleted B*57:03-Tg mice or littermates (LMs), one-way ANOVA with Bonferroni's multiple comparisons correction. Data are a summary of three independent experiments (except the CD4$^+$ T cell-depleted B*57:01-Tg/PD-1$^{−/−}$ mice, serving as the positive control for each independent experiment).

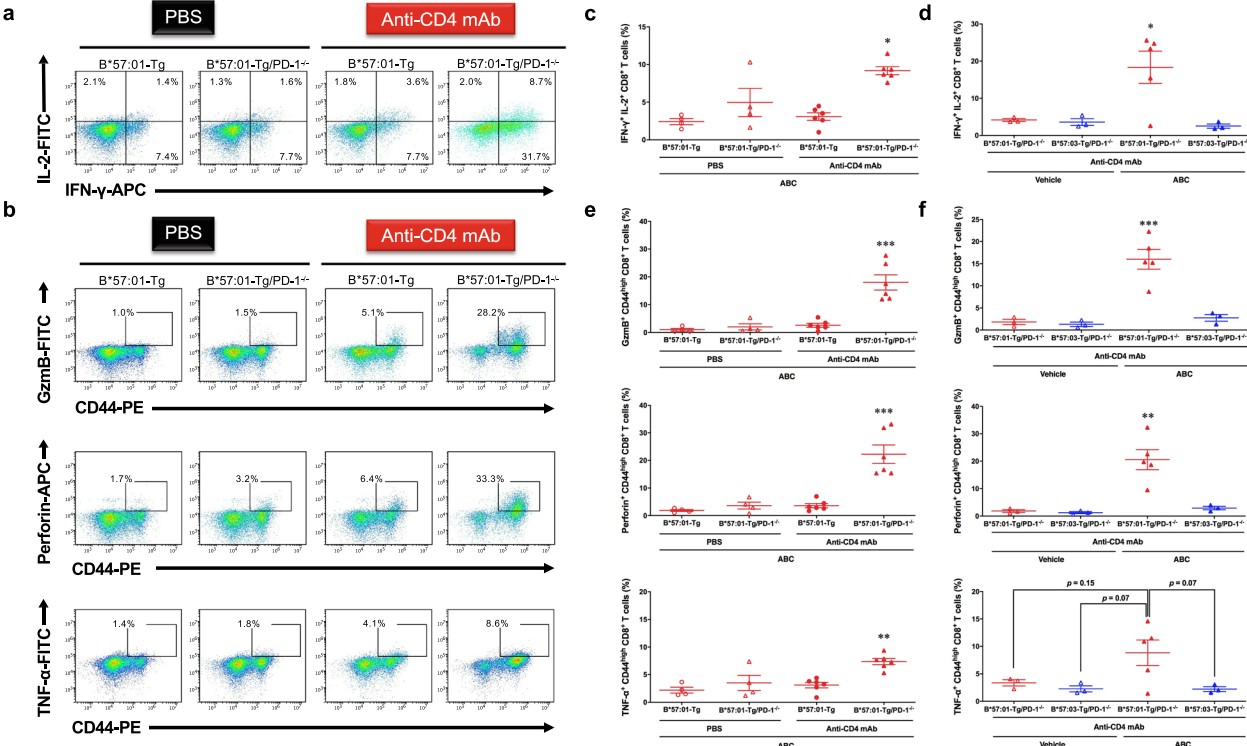

**Fig. 4 Abacavir (ABC)-induced immunotoxicity in CD4⁺ T cell-depleted B*57:01-Tg/PD-1⁻/⁻ transgenic mice. a** and **b** Representative images of photos of the ears or ear section either stained with hematoxylin and eosin (H&E) or immuno-stained with CD8, and nuclear stained. Each group was orally administered (**a**) 1% (w/w) abacavir (ABC) or (**b**) normal diet (vehicle) for 1 week. Each scale bar represents 100 μm. Arrows show the lymphocytic infiltration (H&E staining) and CD8⁺ T cell infiltration (CD8 immunohistochemistry). Data are representative of three independent experiments.

**Fig. 5 Abacavir (ABC)-induced immunotoxicity in CD4⁺ T cell-depleted B*57:01-Tg/PD-1⁻/⁻ transgenic (Tg) mice. a** and **b** Representative dot plots depicting (**a**) cytokine and (**b**) cytolytic granule production in CD8⁺ T cells in the lymph nodes (LNs) derived from CD4⁺ T cell-depleted/PBS-treated B*57:01-Tg mice or B*57:01-Tg/PD-1⁻/⁻ mice. Each group was orally administrated 1% (w/w) abacavir (ABC) for 1 week. Data are representative of four independent experiments. **c–f** Percentage of (**c** and **d**) cytokines and (**e** and **f**) cytolytic granule production in CD8⁺ T cells isolated from the LNs. Each plot represents an individual mouse with the mean ± SEM ($N = 3–6$); *$p < 0.05$, **$p < 0.01$, ***$p < 0.001$ by one-way ANOVA with Bonferroni's multiple comparisons correction, compared with other control groups; indicated $p$ value was obtained from a statistical comparison. Data are a summary of 4 (**c**, **e**) or 3 (**d**, **f**) independent experiments.

We found that the conspicuous toxicity was associated with lymphocytic and CD8⁺ T cell infiltration in the dermic layer of ABC-fed CD4⁺ T cell-depleted B*57:01-Tg/PD-1⁻/⁻ mice (Fig. 4a). In contrast, there was no skin toxicity in CD4⁺ T cell-depleted ABC-fed B*57:03-Tg/PD-1⁻/⁻ mice (Fig. 4a) or vehicle-fed B*57:01-Tg/PD-1⁻/⁻ mice (Fig. 4b).

To examine whether the observed skin toxicity signs in ABC-fed CD4⁺ T cell-depleted B*57:01-Tg/PD-1⁻/⁻ mice were caused by activated cytotoxic T cells, we analyzed the phenotype of CD8⁺ T cells in the LNs (Fig. 5). ABC-treatment increased the secretion of Th1 cytokines (IFN-γ and IL-2) (Fig. 5a) and cytolytic granule (granzyme B (GzmB), perforin, and TNF-α) (Fig. 5b) in CD8⁺ T cells, compared with that in the other control B*57:01-Tg mouse groups. Although ~10% basal population of IFN-γ⁺CD8⁺ T cell was detected in the control groups (Fig. 5a),

it was mostly IL-2 negative. In contrast, ABC-fed CD4⁺ T cell-depleted B*57:01-Tg/PD-1⁻/⁻ mice exhibited a significantly larger population (18.3 ± 4.3%; Fig. 5d) of IFN-γ⁺IL-2⁺CD8⁺ T cells ($p < 0.05$) (Fig. 5c, d). Both IFN-γ⁺ and IL-2⁺CD8⁺ T cells were dominant in the CD44^high activated T cell population (Supplementary Fig. 5a). Moreover, a significant increase in GzmB⁺, perforin⁺, and TNF-α⁺ populations in ABC-fed CD4⁺ T cell-depleted B*57:01-Tg/PD-1⁻/⁻ mice was observed only in activated CD44^highCD8⁺ T cells (Fig. 5b), with mean percentages of 16.0 ± 2.2%, 20.6 ± 3.7%, and 8.8 ± 2.3%, respectively ($p < 0.01$) (Fig. 5e, f). Major populations of GzmB⁺ and perforin⁺ cells in ABC-fed mice were found in IFN-γ⁺CD8⁺ or IL-2⁺CD8⁺ T cells (Supplementary Fig. 5b). These cytotoxic CD8⁺ T cell populations did not significantly differ between ABC-fed and vehicle-fed B*57:03-Tg/PD-1⁻/⁻ mice (Fig. 5d and f).

These observations indicated that ABC treatment of CD4$^+$ T cell-depleted B*57:01-Tg/PD-1$^{-/-}$ mice induced skin hypersensitivity mimicking AHS in humans with the significantly elevated cytotoxic effector memory CD8$^+$ T cells in the LN, which might lead to their infiltration into the dermic layer.

## Discussion

AHS, an ABC-induced IDT with an unpredictable and severe adverse drug effect in clinical scenarios, is associated with fever and rash and may result in death[9]. To better understand the mechanism underlying AHS, several research groups have attempted to reproduce AHS in B*57:01-Tg mice[20,21]. However, these model systems failed to mimic clinically observed AHS, and hence, it is important to consider additional onset risk factors other than the HLA genetic factor. In fact, the introduction of only HLAs was insufficient for mimicking and evaluating IDT in HLA Tg mouse models[19]. From a clinical perspective, AHS predominantly emerges in ABC-prescribed HIV-infected (i.e., CD4$^+$ T cell-deficient) patients[22,24]; therefore, CD4$^+$ T cell depletion should be induced in the B*57:01-Tg mouse model[21]. Moreover, in the present study, PD-1 expression was substantially upregulated in the effector memory CD8$^+$ T cells of ABC-fed B*57:01-Tg mice as well as their CD4$^+$ T cell-depleted counterparts (Fig. 2); this might suppress the induction of skin rash often associated with clinical AHS. Thus, we considered the effect of immune tolerance functions on ABC-induced immunotoxicity to examine whether it determines the susceptibility of B*57:01-Tg-AHS model mice to IDT.

In this study, we created a novel mouse line B*57:01-Tg/PD-1$^{-/-}$, which carries the human–mouse chimeric HLA-B*57:01 and lacks PD-1, and investigated the underlying mechanism of AHS using this model under CD4$^+$ T cell depletion. Oral ABC (1% w/w) treatment for 1 week triggered skin toxicity that could be considered AHS in B*57:01-Tg/PD-1$^{-/-}$ mice due to the infiltration of CD8$^+$ T cells into the dermic layer in the ear and the concurrent occurrence of ear redness (Fig. 4a). We observed systemic LN swelling (Supplementary Fig. 4) with immune-activated effector memory CD8$^+$ T cells occupying half of the total CD8$^+$ T cell population (Fig. 3). Approximately half of these activated effector memory CD8$^+$ T cells exhibited the capacity to produce Th1 cytokines and cytolytic granules (Fig. 5 and Supplementary Fig. 5). In contrast, these phenomena were not observed in ABC-fed CD4$^+$ T cell-depleted B*57:03-Tg/PD-1$^{-/-}$ mice or their vehicle-fed counterparts. These results suggest that the elimination of immune tolerance functions in oral ABC-fed B*57:01-Tg mice induced a significantly increased inflammatory response that resulted in cytotoxic CD8$^+$ T cell production, leading to skin toxicity associated with CD8$^+$ T cell infiltration in the dermic layer.

Cardone et al.[21] reported that the in vivo depletion of CD4$^+$ T cells before ABC administration enhanced dendric cell maturation, which induced systemic ABC-reactive CD8$^+$ T cells with an effector-like and skin-homing phenotype along with CD8$^+$ T cell infiltration and inflammation in drug-sensitized skin. The same tendency was observed in our present study (Supplementary Fig. 1), even after excluding ABC application to the ear (Fig. 1a). Anti-PD-1 mAb treatment itself did not facilitate ABC-CD8$^+$ T cell progression toward a functionally active state with skin-homing potential or predispose ABC-treated B*57:01-Tg mice to develop AHS[21], but the combination of PD-1 knockout and CD4$^+$ T cell depletion in B*57:01-Tg mice had a substantial effect on reproducing AHS (Fig. 4). Impairing immune tolerance by blocking two immune checkpoint proteins, PD-1 and CTLA-4 specifically expressed in CD4$^+$ T$_{reg}$[25], is an effective strategy for reproducing IDT even in non-HLA Tg mice[32,33], and our findings reinforce the importance of eliminating immune tolerance functions to reproduce IDT in HLA Tg mice.

An increase in the basal activation level of CD8$^+$ T cells, achieved by PD-1 knockout or CD4$^+$ T cell depletion, was observed in ABC-fed LMs or vehicle-fed mice (Fig. 3 and Supplementary Fig. 3). This observation might be explained by compensatory regulation, which has been observed in certain in vivo scenarios, including clinical cases[26,34,35]. The increased numbers of effector memory CD8$^+$ cells in the spleen of PD-1$^{-/-}$ mice could be explained by the involvement of PD-1 as a major checkpoint in the central memory (CD44$^{high}$CD62L$^{high}$) of the effector memory phenotype differentiation process, as reported in another study[34]. Similarly, other research groups reported substantially increased percentages of memory CD8$^+$ T cells in association with the CD44$^{high}$CD62L$^{low}$ effector phenotype in CD4$^+$ T cell-depleted mice[35]. In clinical practice, an increased cell population with CD62L$^{low}$CD69$^+$ effector phenotype has been observed in HIV-infected patients[26]. Nevertheless, increasing the proportion of effector memory CD8$^+$ T cells by eliminating critical immune tolerance functions did not affect the expression of a phenotype with cytotoxic potential (Fig. 5 and Supplementary Fig. 5). Thus, our innovative HLA Tg IDT model could provide a clinically relevant immunological profile and be appropriate for studying the underlying mechanism of immune-mediated IDT.

A single cytotoxic T cell clone has been observed in patients with HLA-B*15:02-positive carbamazepine-induced SJS/TEN and in an induced mouse model[36,37]. In contrast, broad polyclonal T cell receptor β (TCRβ) usage was observed using in vitro ABC-stimulated CD8$^+$ T cells from blood[8] or skin of clinical HLA-B*57:01-positive patients[38]. Consistently, the presentation of diverse neo-self-peptides has been detected in ABC-treated HLA-B*57:01-positive antigen-presenting cells[8]. Thus, the polyclonal expansion of cytotoxic CD8$^+$ T cells might be involved in the mechanism underlying HLA-B*57:01-related IDT. We also analyzed the CD8$^+$ T cell receptor (TCR) repertoire in LNs recovered from ABC-fed CD4$^+$ T cell-depleted B*57:01-Tg/PD-1$^{-/-}$ mice (Supplementary Fig. 6). Expectedly, not a single cytotoxic T cell clone but several clones showed more than 1% of Vβ and Jβ usage and combinations of productive sequences in total CD8$^+$ T cells in these mice, inferring TCR-polyclonality. This is consistent with polyclonality observed in the AHS skin biopsies of clinical HLA-B*57:01 positive patients[38]. Therefore, this finding implies that polyclonally expanded CD8$^+$ T cells contributed to the induction of skin hypersensitivity in CD4$^+$ T cell-depleted B*57:01-Tg/PD-1$^{-/-}$ mice, mimicking TCR-polyclonality in AHS. Moreover, this tendency was observed in FLUX-treated cells[39], suggesting that FLUX-induced IDT might be triggered by polyclonal CD8$^+$ T cell clones. The current study demonstrated the potential of CD4$^+$ T cell-depleted B*57:01-Tg/PD-1$^{-/-}$ mice to further examine the IDT-associated mechanism involving AHS. Therefore, future investigations using our model would (i) facilitate the discovery of a universal principle or determining factors that contribute to the occurrence of polyclonal CD8$^+$ T cell-induced HLA-mediated IDT, (ii) predict IDT-causing drugs in the pre-clinical stages, and (iii) find an innovative biomarker for developing more efficient screening tests.

Herein, we demonstrated that PD-1 is involved in preventing AHS induction in chimeric HLA Tg mice. This observation highlighted the effect of PD-1 expression level on the susceptibility to IDT, including AHS. Indeed, single nucleotide polymorphisms (SNPs) in the PD-1 gene, as well as PD-L1 or PD-L2 polymorphisms in humans, determine the risk of developing autoimmune diseases in certain ethnic groups[40]. The rs2227982 polymorphism (missense variant) in the PD-1 gene has been identified as an additional factor in ankylosing spondylitis, which

could not be explained based on the association with HLA-B27 polymorphisms[40–42]. Thus, PD-1 missense SNPs should be similarly considered while assessing the susceptibility to HLA-mediated IDT. However, it is important to note that environmental factors other than genomic alternations and transcriptional mechanisms are involved in regulating PD-1/PD-L1 expression. For example, PD-1 expression is induced by hepatitis B virus (HBV) infection[43], and PD-L1 expression is metabolically controlled by pyruvate kinase M2 in macrophages[44] or upregulated by the stress response to impaired heme production[45]. Furthermore, CTLA-4 SNPs, whose mRNA expression is decreased, are also significantly associated with the risk of autoimmune disease (i.e., Graves' disease) development[46]. In future studies, our B*57:01-Tg mouse model could be used to investigate the association between the expression and function of immune checkpoints and the susceptibility to AHS induction, even if there is a lack of clinical evidence for the relationship between HLA-mediated IDTs, including AHS, and immune tolerance factors. Thus, an HLA Tg mouse model should be useful in determining whether the immune tolerance system is an additional factor affecting the onset risk and severity of HLA-mediated IDTs, which is not limited to conventional factors such as HLA polymorphism.

In conclusion, the current study demonstrated the involvement of the immune tolerance system, including CD4+ T cells and PD-1, in causing AHS in B*57:01-Tg mice. This finding lays a foundation to identify unclarified onset risk factors in HLA-related IDT through further investigation on the molecular regulation of immune checkpoints. We also believe that our study provides a valuable insight to reduce the IDT onset risk, which is an extremely rare and unpredictable adverse event in clinical scenarios, by considering individual differences in immune tolerance levels in humans. Moreover, boosting the immune system in chimeric HLA-B*57:01-Tg mice had a beneficial effect on reproducing AHS induced by oral ABC administration, suggesting that this strategy might be appropriate to achieve higher sensitivity and capability in mimicking IDT in chimeric HLA Tg mouse populations. A promising chimeric HLA Tg mouse model could become a valuable tool for investigating the underlying mechanism of IDT and predicting IDT-causing drugs in the preclinical stages. To date, several attempts have been made to identify potentially hazardous drugs with a risk of causing HLA-mediated IDT by molecule structure-based assessment between HLA and drugs using either an in vitro cell model combined with the phage display method or an in silico computational method[47,48]. In combination with these approaches, our findings will help understand the fundamental mechanism of association between HLA and IDT-causing drugs and the risk of IDT onset in humans.

## Methods

**Materials**. ABC sulfate was purchased from Carbosynth, Ltd. (Compton, Berkshire, UK). RPMI-1640 medium and antibiotic–antimycotic solution were purchased from Nacalai Tesque (Kyoto, Japan). For the FACS analysis, PE-Cy/7 anti-mCD8a antibody (Ab) (53-6.7), PE anti-mouse/humanCD44 Ab (IM7), FITC anti-mCD62L Ab (MEL-14), FITC anti-human/mouse Granzyme B Ab (GB11), APC anti-mIFNγ Ab (XMG1.2), FITC anti-mIL2 Ab (JES6-5H4), APC anti-mPD-1 Ab (RMP1-30), APC anti-mouse Perforin Ab (S16009A), and FITC anti-mouse TNF-α Ab (MP6-XT22) were purchased from BioLegend (San Diego, CA, USA). All other chemicals and solvents were of analytical grade unless otherwise noted.

**Animals**. Tg mice carrying chimeric HLA-B*57:01 or HLA-B*57:03 (B*57:01-Tg and B*57:03-Tg, respectively (in-house colony)) were generated as previously described[20]. To produce HLA Tg/PD-1 knockout mice (B*57:01-Tg/PD-1−/− or B*57:03-Tg/PD-1−/− (in-house colony)), either B*57:01-Tg or B*57:03-Tg mice were mated with PD-1 knockout mice (B6.l29S2-Pdcd1 < tm1Hon > /HonRbrc), provided by RIKEN BRC through the National Bio-Resource Project of the MEXT, Japan[49]. All experiments were conducted using male mice in the C57BL/6 background. PCR-based genotyping was performed using genomic DNA extracted from mouse tails with the following forward/reverse primer pairs: chimeric HLA

forward, 5′-GAGCTACTCTCAGGCTGCGTG-3′, and reverse, 5′-CATGTTAG-CAGACTTCCTCTGCC-3′; Pd-1 forward, 5′-CTCGGCCATGGGACGTAGGG-3′, and reverse, 5′- GGGTCTGCAGCATGCTAATGGCTG-3′. CD4+ T cell depletion was performed as previously reported[21] with a minor modification. The mice received two i.p. injections (3 days before day 0 and on day 1) of 0.25 mg anti-CD4 mAb (clone: GK1.5, BioLegend) or PBS as the vehicle control. For 3 week-fed mouse groups, the mice received three more i.p. injections (on days 8, 15, and 18) of mAb or vehicle. All mice used in this study were 8–16 weeks old. One to two mice per group were used in each independent experiment.

All animal procedures were approved by the Animal Care Committee of Chiba University (Chiba, Japan) (Animal Experiment Protocols: 2-332 and 3-87). The mice were treated humanely in accordance with the guidelines[50] issued by the National Institutes of Health (Bethesda, MD, USA).

**Detection of effector memory CD8+ T cells and assessment of PD-1 expression after ABC oral administration**. The mice were fed either a normal diet or rodent chow containing 1% (w/w) ABC for 1 week. After euthanizing the mice by cervical dislocation, cells were isolated by draining the auricular LN and spleen. Lymphocytes were co-stained with anti-mCD62L Ab, anti-mouse/humanCD44 Ab, anti-mCD8a Ab, and anti-mPD-1 Ab for 30 min at 4 °C. EC800 cell analyzer (SONY, Tokyo, Japan) was used for FACS analysis, and data were analyzed using Flowlogic software (Inivai Technologies, Victoria, Australia). Gating strategies used for flow cytometry are shown in Supplementary Fig. 7.

**H&E staining and IHC**. After feeding the mice with either a normal diet or rodent chow containing ABC (1% w/w) for 1 week or 3 weeks, ear biopsies were collected and embedded in Tissue-Tek® O.C.T. Compound (Sakura Finetek; Tokyo, Japan) and cryopreserved using liquid N2-cold hexane. The tissues were sliced to 5-μm thick sections using a Leica CM3050S cryotome (Leica Biosystems, Wetzlar, Germany). For H&E staining, each section was fixed in 5% formaldehyde for 15 s and stained with hematoxylin (MUTO PURE CHEMICALS CO., LTD., Tokyo, Japan) at 42 °C for 1 min and eosin (MUTO PURE CHEMICALS CO., LTD.) for 1 min. Entellan® (Merck Millipore; Billerica, MA, USA) was used for mounting the stained sections.

CD8 IHC was conducted as reported previously[29]. Briefly, the sections were fixed with acetone for 5 min. Next, the samples were blocked with 5% FBS for 30 min at room temperature (15–25 °C). Each sample was incubated for 1 h at room temperature with rat anti-CD8a Ab (clone: YTS169.4; Abcam, Cambridge, UK), diluted 1:250 in PBS containing 0.1% BSA. The sections were then incubated for 1 h at room temperature with the corresponding Alexa Fluor-conjugated secondary Ab (Abcam) and Hoechst 33342 nuclear stain (Thermo Fisher), diluted 1:250 and 1:1000, respectively, in PBS containing 0.1% BSA. Vectashield (Vector Laboratories, Inc.; Burlingame, CA, USA) was used for mounting the stained sections.

All samples were imaged using a BZ-X700 microscope (Keyence, Osaka, Japan).

**Measurement of T cell cytokine and cytolytic granule production**. To measure cytokine and cytolytic granule production, cells were isolated by draining the auricular, axillary, brachial, cervical, and inguinal LNs (pooled LNs) of mice fed either a normal diet or rodent chow containing 1% (w/w) ABC for 1 week. The cells were seeded in 96-well plates at a density of $5.0 \times 10^5$ cells/well in 100 μL RPMI-1640 medium containing 10% FBS, 2 mM L-glutamine, 1 mM sodium pyruvate, 1× MEM non-essential amino acids (Nacalai Tesque), 10 mM HEPES, 0.05 mM 2-mercaptoethanol, penicillin (100 units/mL), and streptomycin (100 μg/mL). The plates were then incubated at 37 °C in a humidified atmosphere of 95% air/5% CO2 for 4 h by stimulating with Cell Activation Cocktail (phorbol 12-myristate-13-acetate and ionomycin) (BioLegend) and monensin (BioLegend).

For intracellular staining, the cultured cells were subjected to the following treatment. Briefly, the cell surfaces were initially stained with anti-mCD8a Ab and anti-mouse/humanCD44 Ab for 30 min at 4 °C. The cells were then fixed with 4% paraformaldehyde for 10 min at room temperature and permeabilized with 0.5% (v/v) Triton X-100 for 10 min at 4 °C. Finally, all samples were incubated with anti-cytokine Ab or cytolytic granule Ab for 30 min at 4 °C.

**T-cell receptor repertoire analysis**. RNA was extracted from pooled LNs-derived CD8+ T cells ($1.0 \times 10^5$–$10^6$ cells/sample) and mixed with Sepasol RNA I SuperG (Nacalai Tesque) after positive selection using the MACS® Cell Separation Kit (Miltenyi Biotec, Bergisch Gladbach, Germany). Next-generation sequencing for the TCRβ analysis was performed at the Repertoire Genesis Incorporation (Osaka, Japan) using the unbiased gene amplification method with Adaptor-Ligation PCR by Miseq (Illumina, Inc., San Diego, CA, USA).

**Statistics and reproducibility**. Statistical analyses were performed using Graph-Pad Prism 8 Software (GraphPad Software, La Jolla, CA, USA). Significance was determined using Bonferroni's test for multiple comparisons following one-way ANOVA. In all cases, p values of <0.05 were considered statistically significant. All experiments have been replicated at least three times and are reproducible.

**Reporting summary**. Further information on research design is available in the Nature Research Reporting Summary linked to this article.

## Data availability

The source data of data presented in Figs. 1a, 1c, 2b, 3b, 4a, b, and 5c–f are provided as Supplementary Data 1. The sequence data that support the findings of this study have been deposited at the DNA Data Bank of Japan (DDBJ) Sequence Read Archive (DRA) database under accession numbers DRA011282 and DRA011732. All other data are available from the authors upon reasonable request.

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

## Acknowledgements

We express our gratitude to Daiichi Sankyo Company, Ltd. (TaNeDS program) for beneficial suggestions in this study. We would like to thank Editage (www.editage.com) for English language editing. This work was supported by the Japan Society for the Promotion of Science (JSPS) (JSPS KAKENHI Grant Nos. 16K18932, 17J03861, 19H03386, 20K22801, and 21H02640), The Mochida Memorial Foundation for Medical and Pharmaceutical Research, and Takeda Science Foundation.

## Author contributions

T.S., S.A., and K.I. designed the study; T.S., S.K., and B.S. conducted the experiments and acquired the data; T.S. and S.A. analyzed the data; T.S., S.A., and K.I. contributed new reagents or analytic tools; T.S., S.K., A.K., S.A., and K.I. wrote or contributed to the writing of the manuscript. The authorship order among co-first authors was determined randomly.

## Competing interests

The authors declare no competing interests.
