## [Peer Review File · Communications Biology]

Regulation of the immune tolerance system determines the susceptibility to HLA-mediated abacavir-induced skin toxicityReviewers' comments:

Reviewer #1 (Remarks to the Author):

Susukida et al continue the efforts to generate mouse model systems that mimic aspects of the AHS linked to HLA_B*57:01 expression of the human. they employ their previously reported chimeric HLA-B*57:01-Tg animals (with mouse Kd alpha3domain), and pursue a line of investigation geared to reproduce some of the phenomenology of the human B*57:01-linked condition. The work extends previous studies (from the authors and from Cardone et al) which explored B*57-linked hypersensitivity to Abacavir in Tg mouse models. This work explores skin sensitization and is novel for studying oral exposure to Abacavir as well as the effects of using a mouse strain that is also PD-1KO. This the work is logically presented and the data are clear. The writing is a little self-serving, in that some of the earlier work from Cadoneet al relating to the role of CD4 cells and of costimulation is not fully alluded to. Nevertheless the work as presented is novel and adds to the development of more accurate models for AHS.

Reviewer #2 (Remarks to the Author):

Abacavir hypersensitivity syndrome (AHS) is an idiosyncratic adverse drug reaction that is strongly associated with carriage of HLA-B*57:01, occurring in approximately half of HLA-B*57:01 carriers administered abacavir (prior to implementation of HLA screening). HLA-B*57:01-Tg mice represent a promising model for investigation of factors beyond the HLA that contribute to the development of AHS. Here Susukida et al, build on their HLA-B*57:01-Tg mouse model (Susukida et al 2018, ref 20), and observations that CD4+ T cell depletion enhances CD8+ T cell responses to abacavir in HLA-B*57:01+ transgenic mice (Cardone et al 2018, ref 21), to investigate the contribution of PD-1 to abacavir tolerance through generation of HLA-B*57:01-Tg/PD1^{-/-} mice. They demonstrate that PD-1 knock out enhances CD8+ T cell responses beyond CD4+ T cell depletion alone, recapitulating features of AHS. Inclusion of HLA-B*57:03-Tg/PD^{-/-} mice, replicates the HLA specificity seen in humans. Mouse models such as this represent promising avenues for understanding factors beyond the HLA that contribute to drug hypersensitivity reactions, hence are of high interest to the field.

Specific comments:

1. For figure 1a, it is a little unclear from the wording of the first sentence (lines 645-646) of the figure legend if the litter mate control in the image has also received the CD4+ T cell depletion protocol, I assume they have? Could text be added to clarify? Also were control experiments performed here in the Tg mice without CD4+ T cell depletion or without ABC (i.e. ABC alone, or CD4+ depletion alone) as per later experiments?
2. The HLA-B*57:03-Tg mouse speaks to the recapitulation of the HLA specificity of AHS in a mouse model. It first appears in Supplementary figure 1, although it is not referenced in the main text at this point and its utility as a negative control is not currently explained in the manuscript. It would be useful to reference the HLA-B*57:03-Tg data from supplementary fig 1 and include a few lines explanation for readers not familiar with the specificity of abacavir hypersensitivity and the impact of the two amino acid difference between HLA-B*57:01 and HLA-B*57:03 at the end of the paragraph ending at line 105.
3. During the manuscript "vehicle-treated" is most commonly used to mean normal diet without abacavir (eg lines 132 and 151-2), but occasionally used to mean PBS without anti-CD4 mAb (eg. line 102). Whilst "vehicle-fed" (eg line 155) is clearly referring to the diet. For clarity, it might be beneficial to use the term vehicle-fed throughout to avoid potential confusion with the anti-CD4 mAb treatment/PBS vehicle control.
4. Lines 220-221 "therefore, we speculated that CD4+ T cell-depletion should be induced in the B*57:01-Tg mouse model." Given CD4+ T cell depletion has been performed in a previous HLA-B*57:01-Tg mouse model (Cardone et al, reference 21 of the manuscript), it would be useful to acknowledge this here (although it is discussed later at line 243).

5. Line 134 – should this reference supplementary figures 2 and 3 rather than main Fig 2 and 3?

6. Lines 198-200 “However, there were no specific clones in those mice except for no. 1 (CDR3: CASSQGLGENTLYF, TRBV14/TRBJ2-7), illustrating the TCR-polyclonality in CD4+ T cell-depleted B*57:01-Tg/PD-1^{-/-} mice,” – I am not quite clear what is meant here. How is specificity assessed?

Reviewer #3 (Remarks to the Author):

Double transgenic mice with the B*57:01 human polymorphism and that were PD-1 deficient showed greater percentages of CD8+ T cells than littermate controls, and this effect was enhanced in the face of CD4+ T cell depletion. Upon abacavir (ABC) treatment, T cells isolated from the CD4+ T cell-depleted, double transgenic mice showed enhanced expression of IL-2, TNF, granzyme B and perforin than littermate controls with no CD4+ T cell depletion. These mice also had greater rash than littermate controls. The proper experimental design for this study is complex, with multiple factors including B*57:01-Tg, PD-1 deficiency, CD4+ T cell deficiency and ABC treatment. However, most of the data presented in the body of the manuscript involves comparison of double KO, CD 4+ T cell-depleted with littermate controls not CD 4+ T cell depleted. Most of the appropriate controls are presented only as supplemental data, even though they are important to interpreting the factors that drive the success of the animal model. A rigorous experimental design would dictate the various groups being treated at the same time. It is not clear that this was done, so the study potentially lacks rigor in this regard. That said, even the comparison of ABC responses between the double transgenic, CD4+-T cell deficient mice and normal littermate controls is potentially valuable in terms of development of a model that mimics the human ABC-induced, idiosyncratic skin rash. Best would be to rewrite the work including all relevant control data in the body of the manuscript (ie, not just supplemental). Issues also arise with poor presentation of results in the figures, inadequate numbers of replicates

P1, title: The title should be changed to indicate that the study relates specifically to abacavir (ABC) and to dermal toxicity, since abacavir can also damage other tissues. As is, the title is too general.

P3, line 63: “DILI” should be defined.

P3, line 65: “... positive prediction values of each HLA-associated IDT are reported as follows...” The authors should make clear that the values cited relate to skin rash for abacavir and not to other manifestations of abacavir IDT.

P14, line 333: The treatment protocol for abacavir should be described here in detail. Also, the rationale for the dose of ABC used in the study should be stated and defended in light of the dose used in humans.

P5, lines 97-105: This section describes effects of oral ABC combined with ear painting with ABC. The ear painting is irrelevant to any of the subsequent studies described in the manuscript. This section and the associated figure should be omitted.

Fig 1: The legend for Fig 1a indicates “Data are representative of 2 independent experiments.” Does this mean that only 2 mice were used to generate these data? These effects should be quantified and statistical analysis performed. In Fig. 1b, there are no datapoints (dots) for the LM group visible for the flow cytometry analysis.

P6, lines 134-136: “Based on these results, we inferred that a PD-1-mediated immune-tolerance mechanism also contributed to the ABC-induced abnormal immune activation in B*57:01-Tg mice.” The evidence presented for this statement is based on an association between the Tg mice and PD-1 levels; there is no proof that the increase in PD-1 contributed causally to the ear rash.

Fig 4. The “representative” photographs of rash in Fig 4a does not allow one to discern the effects of CD4+ T cell depletion; it only shows the effect of ABC. In Fig 4b, effects are obscured by the

overlays in most cases. In Fig. 4C, the arrow is missing from the merge panel. A major deficiency in this study is that the effects of the knockouts and the CD4+ T cell depletion are presented only as supplemental data. These results are important for interpretation and should be presented in the body of the work. Moreover, it is not clear if these "controls" were performed at the same time, as they should be to be valid controls. Also, it is not clear how many animals had similar results (ie, how many animals were evaluated to come up with these "representative" results? Quantification and use of statistical analysis would render the results more convincing.

P 8, lines 187-190: "These observations indicated that ABC treatment of CD4+ T cell-depleted B*57:01-

Tg/PD-1^{-/-} mice induced skin hypersensitivity via the significantly elevated cytotoxic effector memory CD8+ T cells in the auricular LN and their infiltration into the dermic layer, which mimics AHS in humans." This statement is a bit too strong, since cause and effect was not shown between the CD8+ T cells and the ear rash.

Fig 6: This figure should be deleted. In Fig 6a and 6b, the comparisons are confounded. For example, in Fig 6b double transgenic, CD4+ T cell-depleted mice treated with ABC are compared to single transgenic mice treated with vehicle. This is a confounded comparison from which no valid conclusion can be drawn. Similar confounded comparisons comprise Fig. 6a and 6c.

Responses to the Comment of Reviewer 1:

We thank you for the critical comments and valuable suggestions that have helped to considerably improve our manuscript. We have revised the manuscript accordingly. Our responses to the comments are provided below.

Comment of Reviewer 1:

Susukida et al continue the efforts to generate mouse model systems that mimic aspects of the AHS linked to HLA_B*57:01 expression of the human. they employ their previously reported chimeric HLA-B*57:01-Tg animals (with mouse Kd alpha3domain), and pursue a line of investigation geared to reproduce some of the phenomenology of the human B*57:01-linked condition. The work extends previous studies (from the authors and from Cardone et al) which explored B*57-linked hypersensitivity to Abacavir in Tg mouse models. This work explores skin sensitization and is novel for studying oral exposure to Abacavir as well as the effects of using a mouse strain that is also PD-1KO. This the work is logically presented and the data are clear. The writing is a little self-serving, in that some of the earlier work from Cadone et al relating to the role of CD4 cells and of costimulation is not fully alluded to. Nevertheless the work as presented is novel and adds to the development of more accurate models for AHS.

Response: We thank you for the comment. We have provided information from the study of Cardone et al. in the introduction section as follows:

Page 6, lines 96–101 (revised manuscript): In fact, a previous study suggested the importance of depleting CD4⁺ T cells to exacerbate ABC-induced abnormal immune toxicity by treating another B*57:01-Tg mice line with both i.p. injection and ear painting, possibly by enhancing antigen-presenting cell co-stimulation²¹.

Furthermore, we have cited the study of Cardone et al. (reference 21) in the discussion section as follows:

Page 12, line 266 – Page 13, line 267 (revised manuscript): therefore, CD4⁺ T cell depletion should be induced in the B*57:01-Tg mouse model ²¹.

We hope that our revised manuscript is suitable for publication in *Communications Biology*.

Responses to the Comments of Reviewer 2:

We thank you for the critical comments and valuable suggestions that have helped to considerably improve our manuscript. We have revised the manuscript accordingly. Our responses to the comments are provided below.

Comments of Reviewer 2:

Abacavir hypersensitivity syndrome (AHS) is an idiosyncratic adverse drug reaction that is strongly associated with carriage of HLA-B*57:01, occurring in approximately half of HLA-B*57:01 carriers administered abacavir (prior to implementation of HLA screening). HLA-B*57:01-Tg mice represent a promising model for investigation of factors beyond the HLA that contribute to the development of AHS. Here Susukida et al, build on their HLA-B*57:01-Tg mouse model (Susukida et al 2018, ref 20), and observations that CD4⁺ T cell depletion enhances CD8⁺ T cell responses to abacavir in HLA-B*57:01⁺ transgenic mice (Cardone et al 2018, ref 21), to investigate the contribution of PD-1 to abacavir tolerance through generation of HLA-B*57:01-Tg/PD1^{-/-} mice. They demonstrate that PD-1 knock out enhances CD8⁺ T cell responses beyond CD4⁺ T cell depletion alone, recapitulating features of AHS. Inclusion of HLA-B*57:03-Tg/PD^{-/-} mice, replicates the HLA specificity seen in humans. Mouse models such as this represent promising avenues for understanding factors beyond the HLA that contribute to drug hypersensitivity reactions, hence are of high interest to the field.

Response: We thank you for the valuable comments. We have revised the manuscript accordingly.

Specific comments:

1. For figure 1a, it is a little unclear from the wording of the first sentence (lines 645-646) of the figure legend if the litter mate control in the image has also received the CD4⁺ T cell depletion protocol, I assume they have? Could text be added to clarify? Also were control experiments performed here in the Tg mice without CD4⁺ T cell depletion or without ABC (i.e. ABC alone, or CD4⁺ depletion alone) as per later experiments?

Response: We thank you for pointing this out. Originally, we did not prepare control groups (i.e., ABC alone and CD4⁺ depletion alone); therefore, we re-conducted the experiments by including these controls, with three replicates. The results are presented as new Fig. 1 in the revised manuscript. Furthermore, we have revised the relevant parts in the results section and the corresponding figure legend as follows:

Page 7, lines 136–142 (revised manuscript): CD8⁺ T cell epidermal infiltration in the ear was only observed in the CD4⁺ T cell-depleted ABC-fed B*57:01-Tg mice (Fig. 1a), which displayed a significant increase in the percentage of effector memory CD8⁺ T cells (CD44^{high}CD62L^{low}) in the auricular lymph node (LN) ($p < 0.05$) (Fig. 1b and c). These signs of skin hypersensitivity on the ears were observed on day 21 of oral ABC administration irrespective of whether the ears were painted or not.

Page 32, lines 789–790 (revised manuscript): Both groups received oral ABC (1% w/w) or a normal diet (vehicle) for 3 weeks, with i.p. injections of 0.25 mg anti-CD4 mAb or PBS.

2. The HLA-B*57:03-Tg mouse speaks to the recapitulation of the HLA specificity of AHS in a mouse model. It first appears in Supplementary figure 1, although it is not referenced in the

main text at this point and its utility as a negative control is not currently explained in the manuscript. It would be useful to reference the HLA-B*57:03-Tg data from supplementary fig 1 and include a few lines explanation for readers not familiar with the specificity of abacavir hypersensitivity and the impact of the two amino acid difference between HLA-B*57:01 and HLA-B*57:03 at the end of the paragraph ending at line 105

Response: We appreciate your suggestion. In the main text, we have referred to the HLA-B*57:03-Tg data presented in Supplementary Fig. 1. We have included an explanation for readers not familiar with the specificity of abacavir hypersensitivity and the effect of the two amino acid difference between HLA-B*57:01 and HLA-B*57:03 as follows:

Page 7, lines 124-128 (revised manuscript): Consistent with our previous study finding²⁰, the skin hypersensitivity was not observed even in CD4⁺ T cell-depleted ABC-fed B*57:03-Tg mice (Supplementary Fig. 1), which possess the negative HLA allotype control that ABC cannot specifically react because of a two amino acid difference at the binding site¹⁰.

3. During the manuscript “vehicle-treated” is most commonly used to mean normal diet without abacavir (eg lines 132 and 151-2), but occasionally used to mean PBS without anti-CD4 mAb (eg. line 102). Whilst “vehicle-fed” (eg line 155) is clearly referring to the diet. For clarity, it might be beneficial to use the term vehicle-fed throughout to avoid potential confusion with the anti-CD4 mAb treatment/PBS vehicle control.

Response: We appreciate your suggestion. As you pointed out, we have changed “vehicle-treated” to “vehicle-fed” in the revised manuscript as follows:

Page 7, lines 167–169 (revised manuscript): CD4⁺ T cell-depleted vehicle-fed B*57:01-Tg mice **did not** show a significant increase in PD-1 expression on effector memory CD8⁺ T cells ($p > 0.99$) (Supplementary Fig. 2).

Page 10, lines 197–201 (revised manuscript): Consistent with the increased occurrence of effector memory CD8⁺ T cells, robust signs of ear inflammation (Fig. 4a) and a systemic increase in LN size (Supplementary Fig. 4) were observed in the ABC-fed CD4⁺ T cell-depleted B*57:01-Tg/PD-1^{-/-} mice but not in other tested groups including the vehicle-fed counterparts.

Page 13, lines 283–285 (revised manuscript): In contrast, these phenomena were not observed in ABC-fed CD4⁺ T cell-depleted B*57:03-Tg/PD-1^{-/-} mice or their vehicle-fed counterparts.

Page 14, lines 303–305 (revised manuscript): An increase in the basal activation level of CD8⁺ T cells, achieved by PD-1 knockout or CD4⁺ T cell depletion, was observed in ABC-fed LMs or vehicle-fed mice (Fig. 3 and Supplementary Fig. 3).

4. Lines 220-221 “therefore, we speculated that CD4⁺ T cell-depletion should be induced in the B*57:01-Tg mouse model.” Given CD4⁺ T cell depletion has been performed in a previous HLA-B*57:01-Tg mouse model (Cardone et al, reference 21 of the manuscript), it would be useful to acknowledge this here (although it is discussed later at line 243).

Response: We appreciate your suggestion. Per your suggestion, we have cited the study of Cardone et al. (reference 21) to support the indicated sentence as follows:

Page 12, line 266 – Page 13, line 267 (revised manuscript): therefore, CD4⁺ T cell depletion should be induced in the B*57:01-Tg mouse model²¹.

5. Line 134 – should this reference supplementary figures 2 and 3 rather than main Fig 2 and 3?

Response: We thank you for pointing this out. We have made the necessary revision as follows:

Page 7, lines 167–169 (revised manuscript): CD4⁺ T cell-depleted vehicle-fed B*57:01-Tg mice **did not** show a significant increase in PD-1 expression on effector memory CD8⁺ T cells ($p > 0.99$) (Supplementary Fig. 2).

6. Lines 198-200 “However, there were no specific clones in those mice except for no. 1 (CDR3: CASSQGLGENTLYF, TRBV14/TRBJ2-7), illustrating the TCR-polyclonality in CD4⁺ T cell-depleted B*57:01-Tg/PD-1^{-/-} mice,” – I am not quite clear what is meant here. How is specificity assessed?

Response: We sincerely apologize for the ambiguous expression. The specificity was assessed based on the visual interpretation from pie charts in Fig. 6b, which would be inappropriate to present and discuss in the manuscript. Therefore, we have deleted the indicated sentence and its corresponding figure (Fig. 6b) in the revised manuscript.

Furthermore, we have moved the subsection “*Analysis of CD8⁺ TCR repertoire in CD4⁺ T cell-depleted B*57:01-Tg/PD-1^{-/-} mice receiving oral ABC*” from the results section to the discussion section (page 15, line 319 - page 17, line 377 in the revised manuscript); the data are presented in Supplementary Fig. 6, as suggested by the other reviewer.

We have deleted the corresponding statement from the abstract as follows:

Page 2, lines 35–39 (revised manuscript): ABC treatment increased the proportion of cytokine- and cytolytic granule-secreting effector memory CD8⁺ T cells **in CD4⁺ T cell-depleted B*57:01-Tg/PD-1^{-/-} mice, thereby** inducing skin toxicity with CD8⁺ T cell infiltration, mimicking AHS.

We hope that our revised manuscript is suitable for publication in *Communications Biology*.

Responses to the Comments of reviewer 3:

We thank you for the critical comments and valuable suggestions that have helped to considerably improve our manuscript. We have revised the manuscript accordingly. Our responses to the comments are provided below.

Comments of reviewer 3:

Double transgenic mice with the B*57:01 human polymorphism and that were PD-1 deficient showed greater percentages of CD8⁺ T cells than littermate controls, and this effect was enhanced in the face of CD4⁺ T cell depletion. Upon abacavir (ABC) treatment, T cells isolated from the CD4⁺ T cell-depleted, double transgenic mice showed enhanced expression of IL-2, TNF, granzyme B and perforin than littermate controls with no CD4⁺ T cell depletion. These mice also had greater rash than littermate controls. The proper experimental design for this study is complex, with multiple factors including B*57:01-Tg, PD-1 deficiency, CD4⁺ T cell deficiency and ABC treatment. However, most of the data presented in the body of the manuscript involves comparison of double KO, CD4⁺ T cell-depleted with littermate controls not CD4⁺ T cell depleted. Most of the appropriate controls are presented only as supplemental data, even though they are important to interpreting the factors that drive the success of the animal model. A rigorous experimental design would dictate the various groups being treated at the same time. It is not clear that this was done, so the study potentially lacks rigor in this regard. That said, even the comparison of ABC responses between the double transgenic, CD4⁺ T cell deficient mice and normal littermate controls is potentially valuable in terms of development of a model that mimics the human ABC-induced, idiosyncratic skin rash. Best would be to rewrite

the work including all relevant control data in the body of the manuscript (ie, not just supplemental). Issues also arise with poor presentation of results in the figures, inadequate numbers of replicates

Response: Per your suggestion, we re-conducted (rewrite) the experiments of all data presented in the body of the manuscript including all relevant control data, and replaced the figures with new figures. Therefore, the effects of the knockouts and CD4⁺ T cell depletion (all relevant control data) are presented in the body of the manuscript, instead of presenting in the Supplementary Data.

We performed 3–4 independent experiments for each control with preparing a corresponding positive control to compare the differences (explained in the latter section of this letter regarding Figs. 1 and 4). All samples used here to generate the data are also provided in the “Source Data File.”

P1, title: The title should be changed to indicate that the study relates specifically to abacavir (ABC) and to dermal toxicity, since abacavir can also damage other tissues. As is, the title is too general.

Response: Per your comment, we have changed the title as follows:

Page 1, lines 1–3 (revised manuscript): Regulation of the immune tolerance system determines the susceptibility to HLA-mediated **abacavir-induced skin** toxicity.

P3, line 63: “DILI” should be defined.

Response: The definition of DILI has already been provided in page 4, lines 52-53 (in the revised manuscript).

P3, line 65: "... positive prediction values of each HLA-associated IDT are reported as follows..." The authors should make clear that the values cited relate to skin rash for abacavir and not to other manifestations of abacavir IDT.

Response: Per your suggestion, we have revised the indicated sentence as follows:

Page 5, lines 73–75 (revised manuscript): In fact, inadequate positive prediction values of each HLA-associated IDT **have been** reported as follows: AHS (**a positive result in epicutaneous patch test**) in HLA-B*57:01, 47.9%¹⁵;

P14, line 333: The treatment protocol for abacavir should be described here in detail. Also, the rationale for the dose of ABC used in the study should be stated and defended in light of the dose used in humans.

Response: Per your comment, we have added the rationale for the dose of ABC used in the study in the results section as follows:

Page 7, lines 130–135 (revised manuscript): Therefore, we examined whether the conspicuous skin toxicity could also be induced without ear painting in CD4⁺ T cell-depleted B*57:01-Tg mice receiving oral 1% (w/w) ABC (Fig. 1). **This dose of ABC could sufficiently induce abnormal CD8⁺ T cell activation in the LN and spleen in B*57:01-Tg mice by maintaining approximately 3-fold higher plasma concentrations of ABC than the maximum plasma concentrations (C_{max}) in humans^{20,29}.**

P5, lines 97-105: This section describes effects of oral ABC combined with ear painting with ABC. The ear painting is irrelevant to any of the subsequent studies described in the manuscript. This section and the associated figure should be omitted.

Response: As Cardone et al. investigated the effect of ABC treatment with both i.p. injection and ear painting on the effective development of skin rash in another B*57:01-Tg mice line, we initially modified this protocol to test the effect of oral ABC combined with ear painting of ABC, in order to reproduce AHS in our B*57:01-Tg mice. Indeed, ABC-induced skin rash was clearly observed in our B*57:01-Tg mice. Thereafter, we examined whether the conspicuous skin toxicity could also be induced without ear painting, which Cardone et al. did not demonstrate. Therefore, we think this section and the associated figure should be retained.

To make this point clear, we have added a justification of the ear painting study in the revised manuscript as follows:

Page 6, line 115 – Page 7, line 120 (revised manuscript): To enable a clear assessment of topical skin toxicity, the mice were fed ABC-containing rodent chow (1% w/w) plus ear painted (50 mg/kg/day) and administered either anti-mouse CD4 monoclonal antibody (mAb) (0.25 mg/body) or vehicle (PBS) by i.p. injection for 3 weeks. This was a modified protocol of another toxicological study using B*57:01-Tg mice, treated with both i.p. injection and ear painting²¹.

Fig 1: The legend for Fig 1a indicates “Data are representative of 2 independent experiments.” Does this mean that only 2 mice were used to generate these data? These effects should be quantified and statistical analysis performed. In Fig. 1b, there are no datapoints (dots) for the LM group visible for the flow cytometry analysis.

Response: As some controls were not used in the original study (Fig. 1a, representative of 2 independent experiments) and because the other reviewer suggested to rewrite this part, we re-conducted the experiments with 3 replicates (3 mice) as an independent experiment. All

samples used here to generate the data are provided in the “Source Data File” (also attached below).

Lymphocytic and CD4⁺ T cell epidermal infiltration in the ear was specifically observed in all CD4⁺ T cell-depleted ABC-fed B*57:01-Tg mice. As it is difficult to quantify data from images and perform statistical analysis, the obvious and reproducible data would be convincing enough to prove this effect.

As for Fig. 1b, we have replaced it with new dot plots. We apologize for the inconvenience.

P6, lines 134-136: “Based on these results, we inferred that a PD-1-mediated immune-tolerance mechanism also contributed to the ABC-induced abnormal immune activation in B*57:01-Tg mice.” The evidence presented for this statement is based on an association between the Tg mice and PD-1 levels; there is no proof that the increase in PD-1 contributed causally to the ear rash.

Response: Per your suggestion, we have deleted this statement and revised the next sentence as follows:

Page 9, lines 174–176 (revised manuscript): We further examined whether the upregulation of PD-1 could suppress ABC-induced abnormal immune activation and skin hypersensitivity in B*57:01-Tg mice.

Fig 4. The “representative” photographs of rash in Fig 4a does not allow one to discern the

effects of CD4⁺ T cell depletion; it only shows the effect of ABC. In Fig 4b, effects are obscured by the overlays in most cases. In Fig. 4C, the arrow is missing from the merge panel. A major deficiency in this study is that the effects of the knockouts and the CD4⁺ T cell depletion are presented only as supplemental data. These results are important for interpretation and should be presented in the body of the work. Moreover, it is not clear if these “controls” were performed at the same time, as they should be to be valid controls. Also, it is not clear how many animals had similar results (ie, how many animals were evaluated to come up with these “representative” results? Quantification and use of statistical analysis would render the results more convincing.

Response: We thank you for pointing out the errors in this figure. We re-conducted the experiments related to data presented in Figs. 2–5 and replaced the figures with new figures. All samples used here to generate the data are also provided in the “Source Data File.” We performed 3–4 independent experiments for each control using ABC-fed CD4⁺ T cell-depleted B*57:01-Tg/PD-1^{-/-} mice as a valid positive control to compare the differences. Furthermore, the effects of the knockouts and CD4⁺ T cell depletion (all relevant control data) are presented in the new figures, instead of presenting in the Supplementary Data.

Similar to the data presented Fig. 1, skin rash and lymphocytic/CD8⁺ T cell epidermal infiltration in the ear were clearly and specifically observed in all CD4⁺ T cell-depleted ABC-fed B*57:01-Tg/PD-1^{-/-} mice. As all relevant controls served as valid controls in 3–4 independent experiments, the result is convincing enough.

P 8, lines 187-190: “These observations indicated that ABC treatment of CD4⁺ T cell-depleted B*57:01-Tg/PD-1^{-/-} mice induced skin hypersensitivity via the significantly elevated cytotoxic

effector memory CD8⁺ T cells in the auricular LN and their infiltration into the dermic layer, which mimics AHS in humans.” This statement is a bit too strong, since cause and effect was not shown between the CD8⁺ T cells and the ear rash.

Response: Per your comment, we have changed this sentence in the revised manuscript as follows:

Page 11, lines 232–235 (revised manuscript): These observations indicated that ABC treatment of CD4⁺ T cell-depleted B*57:01-Tg/PD-1^{-/-} mice induced skin hypersensitivity **mimicking AHS in humans with** significantly elevated cytotoxic effector memory CD8⁺ T cells in the auricular LN, **which might lead to** their infiltration into the dermic layer.

Fig 6: This figure should be deleted. In Fig 6a and 6b, the comparisons are confounded. For example, in Fig 6b double transgenic, CD4⁺ T cell-depleted mice treated with ABC are compared to single transgenic mice treated with vehicle. This is a confounded comparison from which no valid conclusion can be drawn. Similar confounded comparisons comprise Fig. 6a and 6c.

Response: Per your comment, we have deleted Fig. 6 from the results section. However, we believe that raising a debate regarding TCR polyclonality in the AHS mice model would be still beneficial to the readers. Thus, we retained the figure indicating the polyclonally expanded CD8⁺ T cells in ABC-fed CD4⁺ T cell-depleted B*57:01-Tg/PD-1^{-/-} mice and alternatively added the supporting information regarding TCR experiments in the discussion section (**page 15, line 319 - page 17, line 377 in the revised manuscript**), which is presented as Supplementary Fig. 6. All compared results were deleted from the manuscript as it was confounded.

We have deleted the corresponding statement from the abstract as follows:

Page 2, lines 35–39 (revised manuscript): ABC treatment increased the proportion of cytokine- and cytolytic granule-secreting effector memory CD8⁺ T cells **in CD4⁺ T cell-depleted B*57:01-Tg/PD-1^{-/-} mice, thereby** inducing skin toxicity with CD8⁺ T cell infiltration, mimicking AHS.

We hope that our revised manuscript is suitable for publication in *Communications Biology*.

REVIEWERS' COMMENTS:

Reviewer #1 (Remarks to the Author):

The revised manuscript describes the response to abacavir in HLA-B*57:01 transgenic mice, with particular attention to the role PD-1, i.e. further immunosuppression, in permitting the visualization of an abacavir dependent immune response. The revised manuscript deals with criticisms made by the reviewers, in particular concerns about control groups, details about the comparison of HLA_B*57:01 and B*57:03, concerns about visibility of control groups. The paper moves somewhat beyond the earlier results of Cardone et al and shows data implicating CD4 depletion and PD-1 deficiency in developing a more applicable mouse model for AHS. The conclusions are certainly novel and the further development of this animal model is worthy of publication.

Reviewer #2 (Remarks to the Author):

My thanks to the authors for diligently addressing previous comments. I acknowledge and credit the significant amount of work performed to do so.

I do however have a few remaining comments for consideration, mainly around clarity. (Line numbers refer to the line in the manuscript with tracked changes).

1. Figures: Many of the figures are very small and have very small labels requiring 400% zoom to be properly read. Is there scope for these to be larger?

2. Lines 343-355: From my understanding the analysis here is not performed on ABC responsive T cells, but on the full CD8 T cell population of the LN. If so, how can the polyclonality of abacavir responsive T cells be evaluated? Furthermore, the figure legend for Sup fig 6 is called "differenced in CD8+ TCR repertoire in pooled LNs of ABC-fed B*57:01-Tg/PD-1^{-/-} mice". Differences from what?

Minor comments:

3. Line 124-128: "negative HLA allotype control" is a little confusing, is "closely related, non-AHS-associated HLA allotype" appropriate?

4. Line 132-135: It would be clearer to include the concentrations

5. Line 143-145: what is the "examined experimental condition", can this be specified?

6. Line 867 (and elsewhere): what is meant by "Data are collapsed"? I assume individual points represent individual animals?

7. Line 928: The source data file for figures 5d-f don't appear to match the panels of the figure correctly, and the headings in the source data file for figures 5c and 5d don't appear to be in the correct places.

Further comments on the source data file:

1. Some of the text labels appear to be corrupted for figs 1 and 4 and need correction. Images of mice are highly pixelated.

2. Fig 2b: not all source data included, should contain data for both the LN and spleen, but only one table

Reviewer #3 (Remarks to the Author):

In their revision, the authors have provided adequate responses to my comments on the original manuscript.

Responses to the Comments of Reviewer 2:

We thank you for the additional critical comments and valuable suggestions that have helped to considerably improve our manuscript further. We have revised the manuscript accordingly. Our responses to the comments are provided below.

Independent of the reviewer's comments, we would like to make the below correction as we misunderstood the experimental condition of figure 5 (*Measurement of T cell cytokine and cytolytic granule production*).

Page 17, line 390 (and elsewhere; revised manuscript): the auricular LN → the **auricular, axillary, brachial, cervical, and inguinal LNs (pooled LNs)**

Comments of Reviewer 2:

My thanks to the authors for diligently addressing previous comments. I acknowledge and credit the significant amount of work performed to do so.

I do however have a few remaining comments for consideration, mainly around clarity. (Line numbers refer to the line in the manuscript with tracked changes).

Response: We thank you for the valuable comments. We have revised the manuscript accordingly.

1. Figures: Many of the figures are very small and have very small labels requiring 400% zoom to be properly read. Is there scope for these to be larger?

Response: We apologize for the inconvenience. Per your comment, we have enlarged all figures in the revised manuscript.

2.Lines 343-355: From my understanding the analysis here is not performed on ABC responsive T cells, but on the full CD8 T cell population of the LN. If so, how can the polyclonality of abacavir responsive T cells be evaluated? Furthermore, the figure legend for Sup fig 6 is called “differenced in CD8⁺ TCR repertoire in pooled LNs of ABC-fed B*57:01-Tg/PD-1^{-/-} mice”. Differences from what?

Response: As you pointed out, the analysis here was performed on the full CD8⁺ T cell population of the LN. According to our data, the percentage of activated CD8⁺ T cells (effector memory T cell) in total CD8⁺ T cells of LN was approximately 65% in ABC-fed CD4⁺ T cell-depleted B*57:01-Tg/PD-1^{-/-} mice (Fig. 3c) while that was around 10% in vehicle-fed condition (Supplementary Fig. 3b), indicating that the ABC-expanded 55% effector memory T cell (i.e., ABC responsive T cells) would be occupying the majority in the analyzed full CD8⁺ T cell population of the LN. Among them, any expanded single CD8⁺ T cell clone showed >10% of frequency (as another study (Ko et al., *J. Allergy Clin. Immunol.* 2011) reported in carbamazepine (CBZ)-stimulated total CD8⁺ T cells from patients with CBZ-SJS/TEN) was not observed in our data. From these rationales, we evaluated the ABC responsive T cell is not composed of a single cytotoxic T cell clone; and therefore, the polyclonal expansion might be inevitably involved. To include this interpretation, we have revised the corresponding sentence in the revised manuscript as follows:

Page 12, lines 267–270 (revised manuscript): Expectedly, **not a single cytotoxic T cell clone** **but** several clones showed more than 1% of V β and J β usage and combinations of productive sequences in total CD8⁺ T cells in these mice, **inferring** TCR-polyclonality.

We also thank you for pointing out the errors in this section. We have deleted “differenced in” from the figure legend title for Supplemental Fig. 6 in the revised version.

Minor comments:

3.Line 124-128: “negative HLA allotype control” is a little confusing, is “closely related, non-AHS-associated HLA allotype” appropriate?

Response: We appreciate your suggestion. Per your comment, we have changed “negative HLA allotype control” to “closely related, non-AHS-associated HLA allotype” in the revised manuscript as follows:

Page 5, lines 106–110 (revised manuscript): Consistent with our previous study finding²⁰, the skin hypersensitivity was not observed even in CD4⁺ T cell-depleted ABC-fed B*57:03-Tg mice (Supplementary Fig. 1), which possess the **closely related, non-AHS-associated HLA allotype** control that ABC cannot specifically react because of a two amino acid difference at the binding site¹⁰.

4.Line 132-135: It would be clearer to include the concentrations

Response: Per your suggestion, we have added the concentration of ABC as follows:

Page 6, lines 114–118 (revised manuscript): This dose of ABC could sufficiently induce abnormal CD8⁺ T cell activation in the LN and spleen in B*57:01-Tg mice by maintaining **an average ABC plasma concentration of 34.5 μM, which is** approximately 3-fold higher than the maximum plasma concentrations (C_{max}) in humans^{20, 29}

5.Line 143-145: what is the “examined experimental condition”, can this be specified?

Response: We sincerely apologize for the ambiguous expression. Per your suggestion, we have revised the indicated term as follows:

Page 6, lines 124–126 (revised manuscript): Moreover, signs of obvious ear inflammation (e.g., redness or congestion) were not observed in the oral ABC administrated CD4⁺ T cell-depleted B*57:01-Tg mice (Fig. 1a).

6.Line 867 (and elsewhere): what is meant by “Data are collapsed”? I assume individual points represent individual animals?

Response: We sincerely apologize for the ambiguous expression. As you pointed out, individual points represent individual animals. To make this point clearer, we have changed “Data are collapsed from individual mice within each group of” to “Data are summary of”, and revised sentences indicating mean values and SEM as “Each plot represents an individual mouse with the mean ± SEM”, in the legends of figures and supplementary figures in the revised manuscript.

7.Line 928: The source data file for figures 5d-f don’t appear to match the panels of the figure correctly, and the headings in the source data file for figures 5c and 5d don’t appear to be in the correct places.

Response: We thank you for pointing out the errors in this section. We have revised the table of figures 5d-f in the source data file. Also, we have revised the corresponding legend in the revised manuscript as follows:

Page 28, lines 700–702 (revised manuscript): Data are summary of 4 (c, e) or 3 (d, f) independent experiments.

Further comments on the source data file:

1. Some of the text labels appear to be corrupted for figs 1 and 4 and need correction. Images of mice are highly pixelated.

Response: We thank you for pointing out the errors in this figure. We have revised the figures 1 and 4 in the source data file.

2. Fig 2b: not all source data included, should contain data for both the LN and spleen, but only one table

Response: We thank you for pointing out the errors in this section. We have revised the table of Fig. 2b in the source data file, including all source data for both the LN and spleen used in the study.

We hope that our revised manuscript is suitable for publication in *Communications Biology*.